# Knowledge and Use of Cervical Cancer Prevention Services among Social Work and Nursing University Students

**DOI:** 10.3390/healthcare10061140

**Published:** 2022-06-19

**Authors:** Maria Moudatsou, Panayiota Vouyiouka, Eleni Karagianni-Hatziskou, Michael Rovithis, Areti Stavropoulou, Sofia Koukouli

**Affiliations:** 1Social Work Department, School of Health Sciences, Hellenic Mediterranean University, GR-71410 Heraklion, Greece; vgiota19@hotmail.com (P.V.); elenaki_k21@hotmail.com (E.K.-H.); koukouli@hmu.gr (S.K.); 2Laboratory of Interdisciplinary Approaches for the Enhancement of Quality of Life, Hellenic Mediterranean University, GR-71410 Heraklion, Greece; rovithis@hmu.gr (M.R.); astavropoulou@uniwa.gr (A.S.); 3Institute of Agri-Food and Life Sciences, Hellenic Mediterranean University Research Centre, Hellenic Mediterranean University, GR-71410 Heraklion, Greece; 4Nursing Department, School of Health Sciences, Hellenic Mediterranean University, GR-71410 Heraklion, Greece; 5Nursing Department, School of Health and Care Sciences, University of West Attica, GR-12243 Athens, Greece

**Keywords:** cervical cancer prevention services, knowledge, social capital, social work students, nursing students, health education

## Abstract

The present study examines: (a) the knowledge of healthcare students on cervical cancer (CC) issues and the use of related preventive services, as well as their association with the field of study and other sociodemographic characteristics; (b) the possible effect of social capital and its parameters. A cross-sectional study was conducted, using a convenience non-probability sampling technique. The final sample consisted of forty-nine social work and fifty-one nursing students. The two groups were similar regarding their sociodemographic characteristics and the knowledge and use of gynecological preventive services. However, the nursing students undertook a PAP smear check-up to a lesser extent (48.6%) compared to social work students (51.4%) (*p* = 0.026). The social capital scores were high for both groups, but social work students were significantly more ‘Tolerant to diversity’. For the total sample, only the ‘Family and friends connections’ subscale correlated with knowledge about the existing gynecological preventive services. Among the main reasons explaining university students’ avoidance of preventive testing were the feelings of fear and embarrassment associated with the PAP smear test. Given the significance of the future professional roles of healthcare students as information sources and leaders in women’s CC preventive behavior, understanding the individual factors contributing to their own adherence is essential. It is equally important to increase their scientific knowledge through the improvement of academic curricula regarding these issues.

## 1. Introduction

Cervical cancer (CC) is the fourth leading cause of cancer death among women worldwide affecting them mainly during their reproductive years [1,2,3]. It is estimated that there were 570,000 new cases and 311,000 deaths of women from cervical cancer globally in 2017 [4] with the 85% of deaths occurring in developing regions [5,6]. Cervical cancer is also ranked as the second most common cancer among women in Europe [7].

It is well known that the human papilloma virus (HPV), the causative agent of cervical cancer, can be detected by the Papanicolaou smear test (PAP test) at an early enough stage to be curable [8] as well as by the liquid-based cytology (LBS) or the HPV test [8,9]. The effectiveness of the PAP test has drastically changed the epidemiological patterns of CC over the recent decades. While HPV vaccination is very promising for the prevention of HPV-related cancers, it is critical that early detection strategies continue to be utilized to achieve cervical cancer elimination [9].

Nevertheless, there is significant variation in the way different countries address the matter of cervical cancer screening [9]. In Greece, women are invited for preventive gynecological examinations and cervical cancer screening either by physicians in Primary Health Care settings or by the gynecologists of the private sector. However, according to a recent study, one third of Greek women were occasionally—not regularly—tested for cervical cancer detection, despite the physicians’ recommendations and the fact that they were aware of the importance of testing [10]. Although non-adherence is a more complex phenomenon, in Greece it is partly due to the lack of a nationally organized population-based screening program [9,11]. The financial crisis of the last decade and the outbreak of the COVID-19 pandemic did not contribute to change this situation.

Preventing gynecological cancers is a multi-factorial process defined by individual, social and cultural parameters [12,13]. Socioeconomic characteristics such as income, education, age, and employment are associated with preventive behaviors [14]. Younger and married women or those living in urban centers are more likely to adhere to gynecological testing procedures as opposed to older, unmarried or those who reside in remote areas [8,14]. Knowledge of cancer prevention issues is also one of the predictors of adherence. Nonetheless, previous studies have shown that there are women who do not comply with cancer prevention procedures despite being aware of their significance for their health [8]. Among other reasons for explaining differences in adherence are subjective susceptibility to cervical cancer or various beliefs about the way tests are perceived [15]. Beliefs related to the benefits and barriers of cervical cancer prevention tests (PAP smear testing, LBS or HPV) have an influence on women’s emotional reactions and, consequently, affect their compliance to them [8]. In addition, the fear of pain or the feeling of discomfort during the procedure, the perception of tests as a threat to virginity, or the fear of diagnosis are additional barriers preventing adherence to the cervical cancer procedures among students [8].

Health professionals in general and community health workers in particular, among them social workers [16,17] and nurses [18], play a pivotal role in health education and health promotion providing relevant information and increasing public awareness on prevention issues [19]. They can intervene in fear management, test’s result distress, treatment of previous traumatic prevention experiences or support in practical matters such as a referral to specialized test centers or scheduling an appointment [18,20,21].

It has been supported by several studies that the personal contact of women with health professionals positively influences their decisions to follow prevention procedures [18,21,22,23]. Face to face or telephone contacts are often beneficial to women’s decisions [22,23,24]. Furthermore, professionals can improve the quality of information they provide on prevention issues and make it accessible even to populations living in poverty [25,26]. It is uncertain, however, whether the professionals themselves or the health profession students know and adequately use the cervical cancer prevention tests.

It has been reported that although health professionals have all the theoretical knowledge needed, they do not often act in accordance with the guidelines deriving from them [27,28]. However, it is vital that health professionals involved in prevention and health education comply with prevention processes themselves so as to be capable of persuading their patients to follow the recommended screening guidelines and treatment [18,28]. Moreover, there is lack of information with regards to the knowledge and compliance levels of university students. In general, they are considered to be more advantaged compared to other social groups, as they have access to specialized information through their education [8,29,30,31,32]. In a sample of 267 female nursing undergraduates in USA, the vast majority of them (84%) had had a PAP smear within the last 3 years, with 81% having had a regular history of obtaining the screening test [8]. Additionally, in a study with university students from different fields, health science students performed better compared to others [33].

Moreover, the intention of women to follow gynecological cancer prevention procedures is directly influenced by the social and cultural environment they live in [34,35,36]. The social environment may either create obstacles or facilitate the testing procedures. When several people in a certain social context follow the health prevention procedures or have a positive view about them, women living in such an environment are subsequently motivated to adopt the same proactive behavior [35]. The social capital, i.e., social interactions and engagement in community activities, can promote individual and collective well-being and positively or negatively influence health [13,35,37,38,39,40,41,42,43]. The mechanisms through which the social capital impacts on health are the dissemination of information as well as the practical and emotional support provided [13,35,36,37].

There is an association between social capital’s components of ‘participation’, ‘social support’, ‘trust’ and ‘reciprocity’, and female cancer prevention [36,44]. In a research study conducted in rural Crete, a positive correlation between individual social capital and knowledge of the correct way to follow prevention procedures was found [13]. Specifically, knowledge of the adequate procedures is associated with social capital’s dimension ‘tolerance of diversity’. Furthermore, following preventive procedures is influenced by the parameters ‘value of life’ and ‘participation in the local community’. Information and the subsequent moral responsibility enhance adherence to health prevention procedures and motivate women to comply with cervical cancer prevention tests [13].

The aim of the present study was twofold: (a) to examine the knowledge and use of the cervical cancer prevention tests and services among social work and nursing students and the effect of the field of studies and other socio-demographic factors; (b) to explore the possible influence of social capital and its components.

## 2. Materials and Methods

### 2.1. Study Design

This was a cross-sectional study conducted in the Nursing and Social Work Departments of the Hellenic Mediterranean University (Crete, Greece).

### 2.2. Population

The population consisted of all female students of the two departments after the completion of their first year of studies.

### 2.3. Sampling

A convenience, non-probability sampling technique was used. The final sample consisted of one hundred female students, forty-nine from the Social Work Department and fifty-one from the Nursing Department.

### 2.4. Measures

For data collection an anonymous questionnaire consisting of three parts was used.

#### 2.4.1. Participants’ Socio-Demographic Profile

The first part included the socio-demographic characteristics of the participants, i.e., gender, age, marital status, field of studies, year of studies, work status, family’s monthly income.

#### 2.4.2. The Social Capital Scale

The second part comprised a social capital measurement. To measure the concept of individual social capital the ‘Social Capital Questionnaire’ (SCQ) was used, created by Onyx and Bullen [45] and consisting of 36 items. The SCQ was translated and validated in Greek (SCQ-G) both psychometrically [46] and cognitively [47]. The Greek version comprises a general social capital factor, as well as six separate components: ‘participation in the local community’ with twelve items (e.g., being an active member of a local organization or club), ‘feelings of safety’ with two items (e.g., feel safe walking after dark), ‘family/friends connections’ with two items (e.g., number of people you talked to yesterday), ‘value of life and social agency’ with twelve items (e.g., get help from friends when in need; local community feels like home), ‘tolerance to diversity’ with two items (e.g., enjoy living among people of different lifestyles). The participants who had a job additionally answered five questions regarding their ‘work connections’ (e.g., workmates are also friends). All items were provided with a 4-point Likert-type scale and higher scores indicate stronger social capital.

#### 2.4.3. Knowledge and Use of Tests

The third part included questions about student’s knowledge of gynecological cancer preventive services, information regarding the cervical cancer prevention test itself, physician’s suggestions regarding the frequency of its use, and the actual uptake of the PAP test. The questions on cervical cancer prevention were extracted from the doctoral thesis of the first author [48].

### 2.5. Data Collection

Prior to the administration of the final questionnaire a pilot study was conducted to confirm that the questions used were comprehensible and that the aim of the study was clear to the study participants. Questionnaires were distributed personally on the university campus by two members of the research team. Participants completed the questionnaire and returned it to the interviewer. During completion the interviewers remained at the disposal of the interviewees to answer possible questions.

### 2.6. Ethical Considerations

Permission for the use of the SCQ in Greek was given by the Health Planning Laboratory of the Medical School of the University of Crete and by Kritsotakis et al. [46]. The research protocol was approved by the Ethics Committee of the Institution to which the senior author is affiliated. Ethical approval was granted by the Faculty of Health before the commencement of the study. Participants were informed about the aim of the study and their informed consent was provided prior to data collection. All questionnaires were anonymous.

### 2.7. Statistical Analyses 

Scores for social capital (total and for each subscale) were estimated by adding individual questions’ scores [46]. Two total scores were calculated in this analysis: one with 31 items (SC31) and a second one with the 36 items (SC36) including for this second score those students who worked at the time of the survey. Moreover, the scores of the different components were also calculated. Descriptive statistics (means, standard deviations, percentages) were used to describe the sample and the main study variables. Continuous variables such as social capital scores were expressed mainly in mean and standard deviation values. In several circumstances, position and dispersion measures were used, such as the median and the lowest–highest values. Categorical variables were expressed in frequency (n) and %. To test the association between discrete variables the Pearson’s chi-squared tests were used. To check if the means between groups were significantly different either an independent sample t-test (for mean differences between two groups) or an analysis of variance (for mean differences among more than two groups) were used. Additionally scatter plots, Box and Whisker plots and bar charts were used for the graphic representation of statistical results. Data entry was performed with an EXCEL for Windows spreadsheet, while statistical analyses were implemented with the IBM SPSS Statistics version 24.0. Statistical significance for all analyses was defined as *p* < 0.05.

## 3. Results

### 3.1. Participants’ Sociodemographic Profile

The average age of the women participants was 22 (±2.2) years ranging from 19–35 years. The two samples were standardized as to their demographic, professional, and financial characteristics. No statistically significant differences were found regarding the year of study, the age of the participants, their family status, and financial, social insurance or professional characteristics. For both groups most participants were in their early 20s (mean age 22.2 years) and the majority in the second and third year of their studies (Table 1).

### 3.2. Knowledge and Use of Gynecological Preventive Services and PAP Smear Test

Most participants reported that they ‘had visited a gynecologist’ (84%). Among those who ‘never visited a gynecologist’ the majority were nursing students (62.5%). However, the difference between the two groups was not statistically significant (*p* = 0.315). In addition, 82.0% of the students stated to have been adequately informed of how a PAP test is performed, while only one student from the Social Work Department reported that she had no information at all about the PAP test (*p* = 0.381) (Table 2). Seventy-six percent of all students reported that a PAP screening should be annually performed, according to their doctor’s suggestion. An interesting finding shows that, among those who never underwent a PAP test, the majority 77.8% (*n* = 14) came from the group of Nursing students (*p* = 0.026) (Table 2).

The principal reason that prevented students from not having done a PAP test was “fear or shame”. Other responses were, “it just hasn’t happened”, or “I am not sexually active”, while only four students considered that “it was not necessary”. With the exception of the category “Other reasons” the rest of the answers had the same distribution for the two samples under study (Figure 1).

From a total of those who responded about the things that should change regarding the prevention procedure, thirteen answered “Nothing should change”, while three of them thought that the prevention tests should be administered free of charge (note: in Greece the PAP test is free of charge only in public hospitals). Additionally, three students stated that women should receive more information on the subject (Figure 2).

### 3.3. Differences in Social Capital Scores between the Two Groups

The descriptive statistics of the social capital scales for the total sample are presented in Table 3. The correlation between students’ demographic, professional, and educational characteristics and social capital was also examined (Table 4). Only the “Tolerance of diversity” component (*p* = 0.004) seems to be significantly associated with the field of study. The Social Work students scored higher (5.9 ± 1.6) compared to the group of Nursing students (5.1 ± 1.2). In all other components and the total social capital scores there were no statistically significant differences between the two groups.

Similar results were found during the analysis of the differences per year of study. In this latter case, the components did not seem to be influenced by year of study and the scales did not show statistically significant differences (*p* > 0.05) with the exemption of the “Tolerance of diversity” subscale (*p* = 0.011) on which the graduate students’ score was the lowest (4.6 ± 1.6) (Table 5).

### 3.4. Social Capital’s Association with Knowledge and Use of Preventive Services

Further examination was performed for potential differentiation in the social capital between the students who had visited a gynecologist and those who had not. Results are depicted in Table 6 according to which a visit to the gynecologist does not correlate with differences in social capital.

In addition, students who stated that they had done at least one PAP test did not statistically differ from the rest in any of the social capital scales under investigation (results not presented here). However, there is statistical difference in the “Family and friends connections” component (*p* = 0.018) of the social capital scale among students who had been informed of the existence of prevention services and those who did not have this information. The students who said that they did not know where to find gynecological cancer prevention services scored higher on the ‘Family and friends connections’ subscale (5.7 ± 1.4) compared to those who had that knowledge (4.8 ± 1.2) (Table 7).

The reasons the students presented for not uptaking a PAP test were compared against social capital scores. There was a general differentiation tendency (*p* = 0.068) for the 31-question scale. There was also a statistical significance for the “Participation in the local community” component *p* < 0.012 (Figure 3).

Examining the group of students who had never had a PAP test due to fear/shame in relation to the rest of the groups, we found a statistical significance in the overall social capital between the former (77.7 ± 6.8) and those who did not share the above feelings (72.6 ± 7.3), (*p* = 0.038) (Figure 4).

## 4. Discussion

The primary aim of the present study was to investigate the knowledge and adherence to cervical cancer prevention services of social work and nursing students in relation to social and demographic factors. A secondary aim was to explore the influence of social capital and its parameters. The study focused on nursing and social work students as the knowledge and attitudes of future health professionals is pivotal in creating a culture of prevention for the women of the general population [49].

### 4.1. Knowledge and Adherence to CC Preventive Test

One of the main findings, corroborating previous results in similar student samples, was that knowledge is closely linked with adherence to the use of preventive services. It is well documented that the lack of knowledge poses a major barrier in cervical cancer prevention among students in various countries [8,15,50]. In a study among female students from Sub-Saharan Africa in the UK universities, participants had both low percentage of knowledge and low compliance to the cervical cancer prevention test [51]. On the other hand, in a study on knowledge and adherence to this test among nursing students in Soudan, it was found that there were both knowledge and a willingness to uptake the test [18]. Similarly, high scores of knowledge and positive attitudes towards the cervical cancer preventive test were found in a study among female university healthcare students in Malaysia [25]. Moreover, in other studies comparing the knowledge of health science students to students from other fields, the former were more aware of these issues [49,52].

Thus, it was expected that being a university student in general and of health sciences in particular would positively affect the levels of adherence, as health and social care academic curricula reinforce health education and promote healthy attitudes and behaviors [49]. Our results confirm those of previous studies as they showed high information levels regarding gynecological preventive tests and knowledge about the existence of gynecological services or the way a PAP test is performed for both groups. However, against our expectations, the non-adherence rates to cervical cancer prevention tests were significantly higher for nursing students compared to social work students. Other researchers in Greece have also found relatively low participation rates in nursing students. According to Trizeli et al. [32] the percentage of nursing students who followed prevention procedures reached 63.1%.

One explanation of the difference between social work and nursing students regarding the adherence rates could be found in their curricula. Through theoretical modules in public health and health education, social work students learn to recognize the importance of the psychosocial dimension in health and prevention issues. Additionally, through the laboratory modules, they have the opportunity to be trained in community health and social care settings that specialize in issues of education and health promotion and mainly in a holistic health intervention in matters of prevention, treatment, and rehabilitation. On the other hand, issues of health promotion and disease prevention are also discussed throughout the nursing curriculum. More specifically, prevention and health promotion are taught within the framework of a variety of subjects, such as Community Nursing and Oncology nursing, but modules have a more biomedical approach and a rather clinical orientation. The results of the present study highlight the necessity to further develop and include modules which are clearly focused upon prevention and health promotion, emphasizing motivation theories in health and in learning.

### 4.2. Association of Socio-Demographic Characteristics to the Use of CC Preventive Services

According to their socio-demographic profile, all participants of our sample were very young females (in their early 20s), unmarried and with comparable mean family (low) income. Young people are supposed to comply more readily to preventive tests because, in general, they are more informed and have a higher educational background compared to older ones. In line with that, Moudatsou et al. [13] found that younger women were more likely to adhere to cervical cancer screening, because they were more aware of prevention issues. Additionally, younger and educated women are more likely to adhere to gynecological tests since they are appropriately informed and encouraged by health professionals [14,34,50]. Having sexual partners and sexual intercourse are enabling factors for students in attending cervical cancer prevention tests [50].

Socioeconomic characteristics such as income, type of residence, and type of household, are also associated to adherence to the cervical cancer prevention test. In the present study, students in both groups belong to lower income families. Having a low individual or family income or residing in low-income areas are correlated negatively with compliance with cervical cancer preventing tests [24,34]. Additionally, the type of residence of the participants constitutes another barrier. If the students are notpermanent residents in the university area, they might not be familiar with health care services located near the campus and, thus, be reluctant to use them [8].

Among the main reasons cited that prevented the participants from adhering to the cervical cancer prevention test were ‘fear and shame’. This finding corroborates the results of previous studies [49]. The embarrassment associated with the visualization of the body throughout the procedure can be the cause for some women of non-attendance to the test. Additionally, the fear of pain, or the anxiety from a possible negative diagnosis can become an impediment for them to take the decision to uptake the test [8,13,32,35,50].

### 4.3. Correlation of Social Capital to the Use of CC Preventive Services

Our results showed that both social work and nursing students scored high on the social capital total scale and its parameters. Scores were higher than the average mean of the scale (>3). Specifically, there was no differentiation between the two groups either in total social capital scores or in most of the subscales with regards to the field or year of study, family status, professional status, and family income. Only the subscale ‘Tolerance of diversity’ was significantly associated with the field and the year of study, i.e., the social work students and the fourth-year students scored higher than the nursing students and the rest of the sample respectively. It was also an interesting finding that for the total sample the tolerance of diversity reached the highest score at the fourth year of studies and then decreased. People who are more tolerant to diversity are generally more open-minded and may accept novelty easier, and thus, may more likely adhere to proactive behaviors such as the uptake of a cervical cancer screening test [13]. In the present study, social work students’ more tolerance to diversity could also explain partially their higher adherence rates to cervical cancer prevention tests compared to nursing students.

For the rest of the characteristics referring to the visits to a gynecologist and the frequency of testing, no difference was observed in social capital scores with the exception of the ‘Family and friends connection’ subscale. This subscale was significantly associated with the information about gynecological prevention services, i.e., those who had more frequent communication with family and friends, were less informed regarding the existence of those services. This may be due to the role that family and friends play in providing information and knowledge about prevention procedures [13,14,24,35].

For example, encouragement from the primary significant others is important for prevention issues [14,51]. It has been found that the mother–daughter relationship may be encouraging for compliance with prevention procedures [14]. The social support provided through family and friends’ networks are positively associated with adherence to the cervical cancer prevention test, especially, among young women [24]. Social support through its emotional and informative dimension might provide a relief from stressful and embarrassing feelings related to the procedure of the cervical cancer prevention test [24]. Women are also more likely to exchange information about gynecological prevention issues when they trust each other, as is the case with members of the family and friends [35,36]. Therefore, if these connections exist and are strong, women rely on them and seek information mainly from these inner circles, although this information may not always be accurate. On the contrary, those who have weaker ‘family and friends’ networks seek information by themselves through the official system of available services. 

To sum up, according to this study, the total social capital and most of its subscales, except the ‘family and friends’ dimension, do not significantly influence the adherence to the cervical cancer preventive test and the use of other gynecological services. Our results contrast those of the study held in rural Crete among women aged 35–75 years old, in which the total social capital and three of its parameters (Participation in the local community, Life values, Tolerance of diversity) seem to positively influence female cancer prevention initiatives [13]. Additionally, in a study in Kenya, among university students, they are more likely to attend cervical prevention tests because they are more educated and socially empowered [46]. In another study in Latinas of the U.S. a positive correlation was found between adherence to the cervical cancer prevention test and social capital [36]. A possible explanation for the different results regarding the impact of social capital on the compliance with the cervical cancer prevention test might be its different impact on health determinants in relation to the social and cultural environment in which it is developed [35,36,37,38].

## 5. Limitations of the Study

This study has several limitations. The information on knowledge and adherence to prevention screenings is based on self-reports of the students themselves and was not retrieved from their medical records. Therefore reliability cannot be demonstrated asself-reports about cancer preventive behaviors lack the accuracy of medical records or other documentation [34]. Additionally, self-reports produce usually higher rates than objective indicators, due to the social-desirability bias, i.e., the tendency of survey respondents to answer questions in a manner that will be viewed favorably by others, which can lead to the over-reporting of “good behaviors”. The fact that the social capital is dependent upon and varies according to the social and cultural environment inside which it develops [36,38] does not allow us to generalize the findings of the present study for other age groups in Greece or internationally. However, the present study can prove useful to university departments of countries with similar social and cultural backgrounds. Additionally, the study’s population included only young university students and this has implications for the generalization of our results to older and less educated women. Finally, a possible limitation is posed by the small sample size of only one Greek university. A bigger and more representative sample including students from different faculties of social and health care departments could have provided us with the opportunity to reach more reliable results and generalize our study.

## 6. Conclusions

Cervical cancer prevention continues to be a multi-factorial question and health policies dealing with prevention issues should take that fact into consideration. In this study, the impact of social capital and its parameters on the knowledge and use of relative prevention services was limited. However, the finding that stronger social capital related to ‘family and friends connections’ correlated with less information about the existent gynecological preventive services, indicated that, in some cases, these networks exert more influence on their members than their field of study even in prevention issues. The family plays a central role in developing and maintaining attitudes, values, and behaviors related to health promotion of its members. Family-centered intervention strategies are thus very important to raise awareness for cervical cancer screening in order to develop and sustain a good screening culture.

Moreover, the role that health professionals (nurses, social workers, etc.) play in prevention is of utmost importance. Professionals who are persuaded of the meaning of prevention not only adhere themselves more easily to tests, but more importantly inform and make other women aware of the significance of such tests and also help them to remove any doubts related to fear, pain, agony, or shame and to adopt healthy behaviors. Therefore, it is essential that their training aims also toward this direction. It is necessary to revise educational material to put more emphasis on prevention topics, both at undergraduate and graduate levels. Similarly, educational seminars and continuing education programs are also recommended. Training should however be performed in a hands-on way, so that students and future professionals may adopt prevention habits principally for their own benefit. Additionally, universities should offer health policy modules aimed at prevention and strengthening their students’ physical and mental health.

Moreover, new ways of transmitting the message for cervical cancer prevention might be used for young female students. They would probably prefer additional information on CC prevention via social media, or in a more experiential way and by an interdisciplinary approach that would alleviate their fears and anxieties and answer their questions with empathy. For instance, to improve knowledge and adherence in female students a health education program could be implemented in the university premises with an interdisciplinary approach and experiential actions.

It is also important to underline that, although most social workers have theoretical and practical training in health education and prevention, their active involvement in this area remains limited. The typical channels of transmission of health education messages concerning the prevention of gynecological cancers are the professionals of primary health care and the gynecologists and midwives in the private sector. Social workers could, for example, deal more effectively with the management of emotions and issues related to the psychosocial dimension of prevention (fears, anxieties, shame) and with the support and empowerment of women to participate in cervical cancer screening. In particular, inform women and especially discuss with them about the reasons that prevent them from uptaking gynecological examinations. In parallel, they could connect women with primary health care settings where gynecological examinations are performed or organize informal care networks (friends, acquaintances, neighbors) to facilitate women’s access to and use of preventive services. In addition, the social worker at community level can raise awareness and inform on prevention issues either by a person-to-person approach or by organizing various public events with that aim.

To sum up, the promotion of cervical cancer screening is a multilevel process that should: (a) target women and their families; (b) promote preventive care with current, local health care services, (c) motivate individual behavior change with community-based strategies. To achieve the above however, astronger support and financing of primary care by the state is needed.

## Figures and Tables

**Figure 1 healthcare-10-01140-f001:**
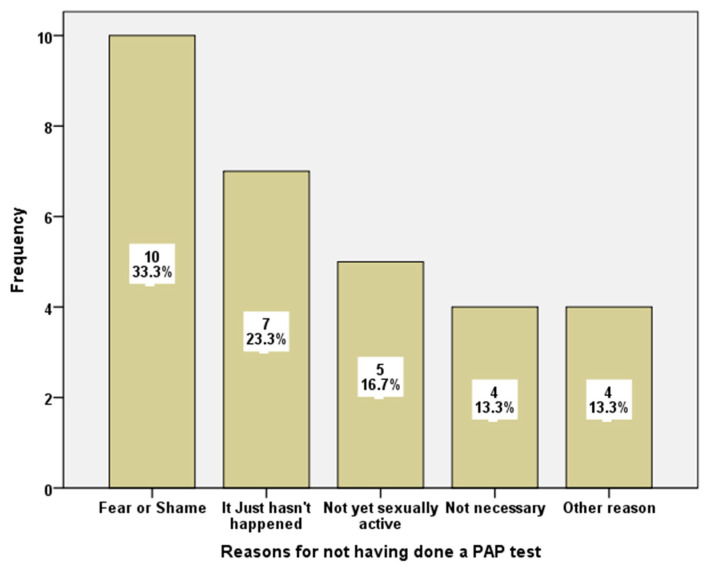
Reasons for not having done a PAP test.

**Figure 2 healthcare-10-01140-f002:**
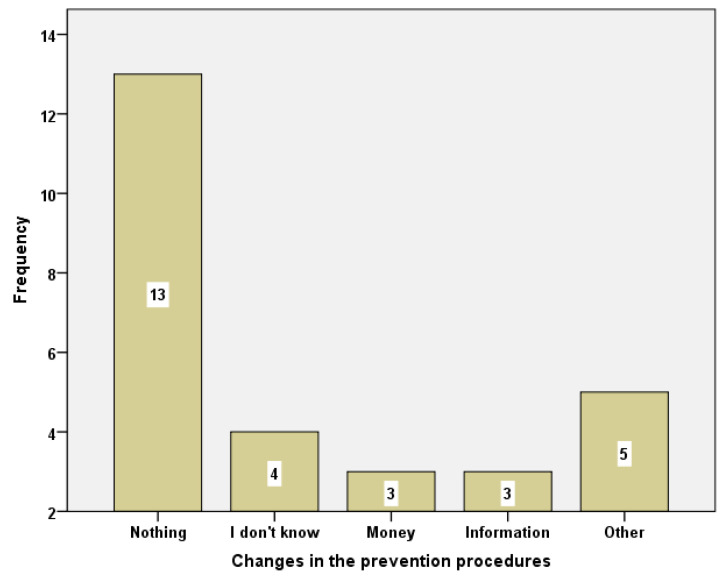
Changes in the prevention procedures.

**Figure 3 healthcare-10-01140-f003:**
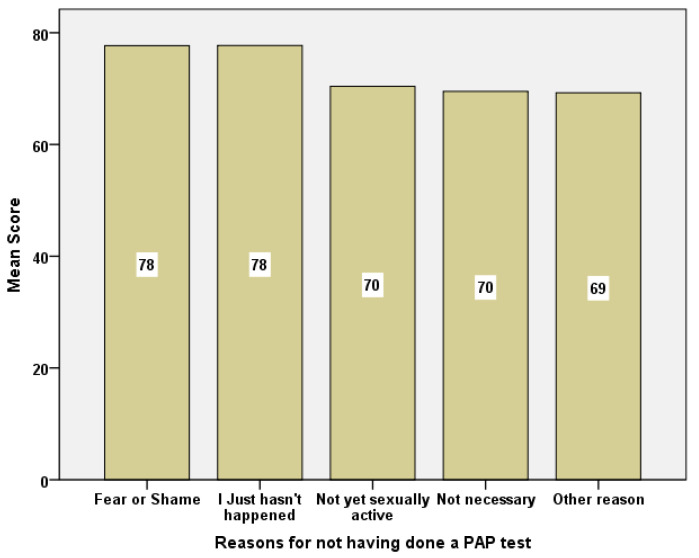
Barchart of social capital mean scores in relation to reasons for not having done a PAP test.

**Figure 4 healthcare-10-01140-f004:**
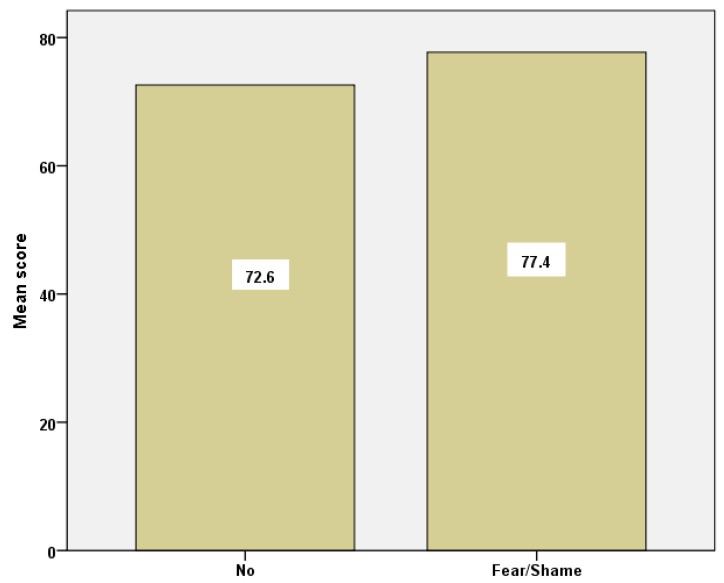
Barchart of social capital mean scores for the group of students who reported ‘fear/shame’ and those who did not.

**Table 1 healthcare-10-01140-t001:** Sociodemographic characteristics of the sample by field of study.

Sociodemographic Variables		Field of Study				
	Variables’ Categories	Social Work (*n* = 49)		Nursing (*n* = 51)		*p*
		*n*	%	*n*	%	
Year of Study	2nd (*n* = 44)	20	45.5	24	54.5	0.138
	3rd (*n* = 31)	12	38.7	19	61.3	
	4th (*n* = 5)	4	80.0	1	20.0	
	>4th (*n* = 20)	13	65.0	7	35.0	
Marital Status	Single (*n* = 99)	45	91.8	44	86.3	0.374
	In relation (*n* = 11)	4	8.2	7	13.7	
Family Income (per month) (€)	<500 (*n* = 72)	35	71.4	37	72.5	0.699
	500–1000 (*n* = 20)	9	18.4	11	21.6	
	>1000 (*n* = 8)	5	10.2	3	5.9	
Age (years)	Mean (SD)	22.2 (2.0)		22.2 (2.0)		0.502

**Table 2 healthcare-10-01140-t002:** Knowledge about cervical cancer and use of gynecological cancer services.

Knowledge and Use Variables	Variables’ Categories	Field of Study		
		Social	Nursing	
		Work		
		*n* (%)	*n* (%)	*p*
Have you ever visited a gynecologist?	No (*n* = 16)	6 (37.5)	10 (62.5)	0.315
	Yes (84)	43 (51.2)	41 (48.8)	
Have you been informed about how a PAP test is performed?	Not at all (*n* = 1)	1 (100)	0 (0.0)	0.381
	A little (*n* = 17)	10 (58.8)	7 (41.2)	
	A lot (*n* = 82)	38 (46.3)	44 (53.7)	
How often does your doctor suggest a PAP test (in years)	1/2 (*n* = 3)	2 (66.7)	1 (33.3)	0.621
	1 (*n* = 76)	34 (44.7)	42 (55.3)	
	2 (*n* = 5)	3 (60.0)	2 (40.0)	
Knowledge of existing gynecological cancer preventive services?	No (*n* = 12)	8 (66.7)	4 (33.3)	0.192
	Yes (*n* = 88)	41 (46.6)	47 (53.4)	
Have you ever done a PAP test?	No (*n* = 18)	4 (22.2)	14 (77.8)	**0.026**
	Yes (*n* = 70)	36 (51.4)	34 (48.6)	

**Table 3 healthcare-10-01140-t003:** Descriptive statistics of the social capital scales for the total sample.

Social Capital Total Scales and Subscales	No of Items	Mean	95%CI	Min	Max
SC-31 *	31	73.1	71.6–74.6	50.0	89.0
SC-36 **	36	75.7	73.8–77.7	50.0	104.0
Value of life	12	34.6	33.8–35.5	25.0	44.0
Participation in the local community	12	20.7	19.9–21.6	13.0	31.0
Feelings of safety	2	5.8	5.5–6.0	2.0	8.0
Family and friend connections	2	4.9	4.6–5.1	2.0	8.0
Tolerance of diversity	2	5.5	5.2–5.8	2.0	8.0
Work connections	3	14.4	12.8–16.1	6.0	20.0

* SC-31 = social capital scale with 31 items, ** SC-36 = social capital scale with 36 items.

**Table 4 healthcare-10-01140-t004:** Social capital differences between fields of study.

Social Capital Total Scales and Subscales	Field of Study				
	Social Work		Nursing		
	Mean SD		Mean SD		*p*
SC-31 *	73.4	7.9	72.8	6.9	0.656
SC-36 **	76.4	10.2	75.1	9.5	0.516
Value of Life	34.8	4.6	34.4	4.2	0.644
Participation in the local community	20.2	3.8	21.2	4.6	0.266
Feelings of safety	5.7	1.3	5.8	1.1	0.485
Family and friend connections	5.0	1.2	4.8	1.3	0.346
Tolerance of diversity	5.9	1.6	5.1	1.2	**0.004**
Work connections	14.3	4.0	14.6	2.6	0.846

* SC-31 = social capital total scale with 31 items, ** SC-36 = social capital total scale with 36 items.

**Table 5 healthcare-10-01140-t005:** Social capital scores by year of studies (mean, sd).

T Social Capital Total Scales and Subscales	Year of Studies				
	2nd Year	3rd Year	4th Year	>4th Year	*p*
SC-31 *	74.0 (6.8)	72.5 (6.7)	79.2 (7.7)	70.7 (8.9)	0.091
SC-36 **	76.5 (8.7)	73.8 (8.8)	82.6 (12.2)	75.2 (12.8)	0.274
Value of life	34.9 (4.8)	34.1 (3.8)	36.2 (4.4)	34.6 (4.3)	0.747
Participation in the local community	20.7 (4.2)	20.8 (4.2)	24.6 (3.0)	19.8 (4.4)	0.156
Feelings of safety	6.0 (1.0)	5.5 (1.0)	5.4 (2.1)	5.7 (1.5)	0.173
Family and friend connections	4.9 (1.4)	4.9 (1.0)	5.4 (1.7)	4.8 (1.1)	0.782
Tolerance of diversity	5.8 (1.3)	5.5 (1.3)	6.2 (2.5)	4.6 (1.6)	**0.011**
Work connections	13.9 (4.5)	14.0 (2.6)	17.0 (-)	15.0 (2.2)	0.826

* SC-31 = social capital total scale with 31 items, ** SC-36 = social capital total scale with 36 items.

**Table 6 healthcare-10-01140-t006:** Social capital scores (means, SDs) and use of a gynecologist.

Social Capital Total Scales and Subscales	Have You Ever Visited a Gynecologist?		
	No	Yes	*p*
SC-31 *	73.7 (7.8)	73.0 (7.4)	0.735
SC-36 **	75.6 (7.7)	75.7 (10.3)	0.948
Value of life	35.3 (4.6)	34.5 (4.3)	0.537
Participation in the local community	19.8 (4.0)	20.9 (4.3)	0.350
Feelings of safety	6.1 (1.1)	5.7 (1.2)	0.188
Family and friend connections	5.1 (1.6)	4.8 (1.2)	0.524
Tolerance of diversity	5.6 (1.8)	5.5 (1.4)	0.736
Work connections	15.0 (1.4)	14.4 (3.6)	0.813

* SC-31 = social capital total scale with 31 items, ** SC-36 = social capital total scale with 36 items.

**Table 7 healthcare-10-01140-t007:** Mean differences in social capital and its components between the students who had and those who had not been informed of prevention services.

T Social Capital Total Scales and Subscales	Gynecological Cancer Prevention Services		
	No	Yes	*p*
SC-31 *	73.6 (5.6)	73.0 (7.6)	0.815
SC-36 **	75.9 (5.9)	75.7 (10.3)	0.939
Value of life	35.9 (4.0)	34.5 (4.4)	0.277
Participation in the local community	19.4 (3.8)	20.9 (4.3)	0.257
Feelings of safety	5.5 (0.7)	5.8 (1.3)	0.429
Family and friends connections	5.7 (1.4)	4.8 (1.2)	**0.018**
Tolerance of diversity	5.6 (1.3)	5.5 (1.5)	0.856
Work connections	14.0 (2.8)	14.5 (3.5)	0.85

* SC-31 = social capital total scale with 31 items, ** SC-36 = social capital total scale with 36 items.

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
