# Peer review of "Knowledge and Use of Cervical Cancer Prevention Services among Social Work and Nursing University Students"

_healthcare, 2022, doi:10.3390/healthcare10061140_

Round 1

Reviewer 1 Report

This paper addresses the differences between social work and nursing female students as it concerns knowledge in managing and screening of CC patients. The paper addresses the adherence of students in adopting what they learn at the university in their own life and the findings show that about half of them do so, and that students in the social sciences are more aware of the social diversity which is related to the behaioural characteristics of each woman. The study shows some gaps that exist in the curricula, and as such I would expect the authors to propose some at least basic changes in the curricula (e.g. inclusion of courses measuring behavior, fear, and addressing specific risk factors for CC such as nicotine (smoking), alcohol, sexyual life style. I would expect the authors to address these issues at least at the discussion level more thoroughly.

In table 1 replace the term 'πτυχιο' with >4th year.

Reviewer 2 Report

Dear Authors,

Thank you for the opportunity to review the manuscript titled, "Knowledge and use of cervical cancer prevention services among Social Work and Nursing University students." I offer the following recommendations for your consideration:

1. Describe the usual communication of cervical cancer screening prevention among the population. For example, what media is used to communicate health education in this population. Does this population prefer to receive information via social media? What health organizations are developing this prevention messaging.

2. Describe specifically how the survey was implemented (i.e., in classes or outside as people walked by on campus).

3. Attach the survey tool as an Appendix to the manuscript so others who may want to replicate or modify it may do so.

4. Describe the role of social workers in health communication. This is not a widespread model in many countries so narrative should be included to describe typical venues/modes of health communication for this population.

5. The finding of the minimal role family and friends play in communicating cervical cancer prevention was surprising. Please provide narrative recommending how this important source of health information can be improved.

Reviewer 3 Report

The authors present a paper that examines cervical cancer knowledge and the usage of related preventive services in a population of healthcare and nursing students. Additionally, the paper aims to associate with the field of study and other sociodemographic characteristics and the possible effect of social capital and its parameters. The authors presented the work in a nice, orderly manner. I present a list of comments for the work. At the same time, I believe that it is worth publishing with minor corrections.

1.     The work is limited by a small number of the research group. Maybe in the future, it will be possible to expand the study group or to see how the image of student groups has changed over the years. 2.     The authors presented that public health students have better knowledge than nursing students. They then showed that it was due to their “tolerance of diversity”. Maybe there is another explanation for the difference in knowledge. 3.     It is worth presenting an action for the future, how to improve knowledge so that the work is not only informative but also educational. Students of both faculties have a real influence on the further education of the society as a role-model. 4.     In work, knowledge about cervical cancer prevention in the context of Pap smear is marked everywhere. However, nothing is written about neither liquid based cytology (LBS) nor the HPV test, which are currently most recommended by gynaecological societies. 5.     In the introduction, "cervical cancer prevention tests" are listed. It is worth mentioning which ones 6.     The introduction states that younger and married women are more likely to visit a gynaecologist than older and unmarried women. Then, added “living in the mountains”. What is the relationship between living in the mountains? It is not written compared to lowland / coastal areas 7.     Please standardize the form of notation of p- p-value in the whole text 8.     Table 1- does social work have to be bold? Nursing is not 9.     Table 1- Greek text crept in – please translate it into English 10.  Table 1- last row - data is shuffled 11.  There are double spaces in the text; edit this please 12.  Figure 1- please improve the quality of axis descriptions; the text is blurry 13.  Figure 1- description "Pap test" - please use a uniform nomenclature - different text in the body, different in the Figure description, different in the axis description 14.  Description of the results from Figure 2 - why the repetition of "13" and the notation “thirteen” 15.  Figure 2- please improve the quality of axis descriptions; the text is blurry 16.  Table 3- "no of items" - please provide uniform nomenclature or description of abbreviations 17.  Table 4- please align the text in the lines 18.  No reference in the text to Table 5 19.  After Figure 2 is Figure 4. Is Figure 3 not pasted? 20.  In references- repeated cited numbers

21.  Three self-citations - it may be worth reducing the amount

Round 2

Reviewer 1 Report

I think the authors improved their MS based on the revisions suggested.